# Strong modulation of second-harmonic generation with very large contrast in semiconducting CdS via high-field domain

Ming-Liang Ren[1], Jacob S. Berger[1], Wenjing Liu[1], Gerui Liu[1] & Ritesh Agarwal[1]

Dynamic control of nonlinear signals is critical for a wide variety of optoelectronic applications, such as signal processing for optical computing. However, controlling nonlinear optical signals with large modulation strengths and near-perfect contrast remains a challenging problem due to intrinsic second-order nonlinear coefficients via bulk or surface contributions. Here, via electrical control, we turn on and tune second-order nonlinear coefficients in semiconducting CdS nanobelts from zero to up to 151 pm V$^{-1}$, a value higher than other intrinsic nonlinear coefficients in CdS. We also observe ultrahigh ON/OFF ratio of >10$^4$ and modulation strengths ~200% V$^{-1}$ of the nonlinear signal. The unusual nonlinear behavior, including super-quadratic voltage and power dependence, is ascribed to the high-field domain, which can be further controlled by near-infrared optical excitation and electrical gating. The ability to electrically control nonlinear optical signals in nanostructures can enable optoelectronic devices such as optical transistors and modulators for on-chip integrated photonics.

[1] Department of Materials Science and Engineering, University of Pennsylvania, Philadelphia, PA 19104, USA. Ming-Liang Ren and Jacob S. Berger contributed equally to this work. Correspondence and requests for materials should be addressed to R.A. (email: riteshag@seas.upenn.edu)

nterest in electrically controlled optical functionalities have been motivated by applications in linear and nonlinear optical systems[1–3], and for fundamental understanding of the electronic and structural symmetry and light-matter interaction in materials[4]. Two most commonly studied phenomena for controlling optical signals via applied electric fields are: electro-optical effect[3] and electric field-induced second-harmonic generation (EFISH)[5–7]. EFISH was first observed in a bulk calcite crystal in 1962[6], in which a strong electric field ($F_l$) was applied to break structural inversion symmetry and induce effective second-order nonlinear coefficients $\chi_{ijk}^{(2)}(2\omega; \omega, \omega, 0)$ by interacting with third-order nonlinear coefficients $\chi_{ijkl}^{(3)}(2\omega; \omega, \omega, 0)$. The effective second-order tensor can be written as, $\chi_{ijk}^{(2)}(2\omega; \omega, \omega, 0) = \chi_{ijk}^{(2)}(2\omega; \omega, \omega) + \chi_{ijkl}^{(3)}(2\omega; \omega, \omega, 0)F_l$, where $\chi_{ijk}^{(2)}(2\omega; \omega, \omega)$ are the intrinsic second-order nonlinear coefficients and correspond to a conventional second-harmonic generation (SHG) process, while $\chi_{ijkl}^{(3)}(2\omega; \omega, \omega, 0)F_l$ represents the d.c. field-induced second-order nonlinear coefficient. The symbol 0 in $\chi_{ijk}^{(2)}(2\omega; \omega, \omega, 0)$ and $\chi_{ijkl}^{(3)}(2\omega; \omega, \omega, 0)$ denotes the d.c. field ($\omega = 0$). EFISH has been utilized in a variety of systems to enhance and modulate SHG signals, such as in silicon/electrolyte[7] and metal–semiconductor[8] interfaces, metal-oxide-semiconductors[9], polymer-filled plasmonic nanoslits[10], and metamaterials[11]. For most reports, the initial SHG signal at zero applied voltage was not zero and only intrinsic non-zero second-order nonlinear coefficients were typically modified via the electric field. Typically EFISH is a weak effect due to the small third-order nonlinear coefficients, and consequently the field-induced second-order nonlinear coefficients are small ($<10$ pm V$^{-1}$)[6], and optical modulation is weak ($<10\%$ V$^{-1}$) with a small tuning range due to the presence of large intrinsic second-order nonlinear coefficients ($\sim 10$ pm V$^{-1}$)[10,11]. Even if a strong electric field is applied to an insulating bulk material or an interface such as semiconductor/electrolyte, it is challenging to switch bulk second-order nonlinear coefficients from zero to very-large values (e.g., $>100$ pm V$^{-1}$), let alone in semiconductors, which typically have a large density of free carriers. It would be desirable if new nonlinear coefficients could be obtained in semiconductors with perfect contrast and strong modulation strengths, via controlling the unique electric field profile, to enable their use for practical applications and integration with other on-chip electronic/photonic devices.

In this paper, we report extremely large tunability of an intrinsically zero second-order nonlinear coefficient ($\chi_{xxx}^{(2)}$, or $d_{11}$) in individual semiconducting CdS nanobelts to a value as high as 151 pm V$^{-1}$ (approximately two-times larger than other intrinsic tensor elements of CdS) by applying a symmetry-breaking electric field. The field-induced second-order nonlinear coefficients, in contrast to intrinsically non-zero coefficients, are intrinsically zero and are produced by applying an external field, which breaks certain relevant mirror symmetries along the field direction. The voltage-dependence of the SHG signal corresponding to $d_{11}$ ($d_{11}$-SHG) exhibits very-high modulation strengths of $\sim 200\%$ V$^{-1}$, and, importantly, with an error function-like shape (rapid rise followed by saturation), which is different from the linear or quadratic response typically observed in EFISH[6,7,10,11] and important for device applications, such as nonlinear optical transistors. These unusual properties are attributed to the formation of a high-field domain in the material, which initiates at the electrical contacts and the domain wall moves with the applied field leading to very-high optical nonlinearities to turn on tensor elements that are otherwise zero. The high-field domain

can be manipulated by infrared (IR) excitation and gate voltage, which further allows for SHG signal modulation in addition to applied electric fields.

## Results

**Observation of $d_{11}$-SHG with applied voltage.** To study dynamically controlled SHG, we fabricated two-terminal devices on a single-CdS nanobelt with a d.c. electric field applied along the CdS $a$-axis (defined as $x$-axis in Fig. 1a), with optical excitation via the fundamental wave (FW) polarized along the CdS $x$-axis, with crystallographic axes confirmed via SHG polarimetry (Supplementary Fig. 1) (see section Methods)[12]. The dimensions ($L \times W \times H$) of the sample in Fig. 1a are $8.2 \times 2.43 \times 0.45$ μm$^3$. Asymmetric contacts (Ti/Au for Ohmic and Au/Al for Schottky) were fabricated in order to apply strong electric field to the sample via the Schottky field (see section Methods) at one junction. A femtosecond pulsed Ti:sapphire laser, tuned from 680 to 1080 nm with $\sim 140$ fs pulse width and 80 MHz repetition rate was focused to a spot size of $\sim 3$ μm to perform the SHG measurements. Wurtzite CdS has mirror symmetry along its $a$-axis with a uniform distribution of its electron wave function without a net nonlinear dipole moment (Fig. 1b), hence corresponding second-order nonlinear coefficients, e.g., $d_{11}$ and $d_{13}$, are zero and produce no SHG signal (Fig. 1c). However, at $V_{DS} = 60$ V bias applied between the source (S) and drain (D) electrodes (Fig. 1c, inset), a very-strong $d_{11}$-SHG signal that corresponds to $d_{11}$ appears near the $S$-electrode (discussed later) at a FW wavelength of 1018 nm. The external normalized conversion efficiency of $d_{11}$-SHG signal ($\eta_{2\omega} = P_{2\omega}/P_\omega^2 = 4.4 \times 10^{-9}$ W$^{-1}$) is 3.76 times stronger than $d_{33}$-SHG at zero bias, an intrinsic and largest second-order nonlinear coefficient of CdS (Fig. 1c). The strong $d_{11}$-SHG signal at 60 V bias indicates that the d.c. field breaks mirror symmetry along the $x$-axis by distorting the electron distribution (Fig. 1b) and induces a net nonlinear dipole moment. By noting $d_{33} = 78$ pm V$^{-1}$[13], the field-induced second-order nonlinear coefficient, $d_{11} = 1.94 \times d_{33}$ $= 151$ pm V$^{-1}$ at $V_{DS} = 60$ V (measured at a fundamental wavelength of 1018 nm), is significantly larger than any other intrinsic second-order nonlinear coefficients of CdS and also other commonly used nonlinear crystals (e.g., LiNbO$_3$ with 27.2 pm V$^{-1}$). The SHG signal ON/OFF ratio is estimated to be $>10^4$, with the ON state corresponding to the saturation SHG (SHG$_{max}$) ($\sim 57$ V in Fig. 1c) and the OFF state being the SHG intensity (SHG$_{min}$) at $\sim 22$ V, where SHG starts to appear and can be reliably recorded above the noise floor (see Supplementary Note 6 for more details). SHG signal contrast, defined as (SHG$_{max}$–SHG$_{min}$)/(SHG$_{max}$+SHG$_{min}$) is $\sim 1$ due to the large ON/OFF ratio.

**Voltage-dependence and wavelength-dependence of $d_{11}$-SHG.** In order to study the dynamic properties of $d_{11}$-SHG, we measured the $d_{11}$-SHG signal with respect to applied voltage at different fundamental wavelengths (1018, 1028, and 1043 nm) (Fig. 2a) and simultaneously measured current–voltage ($I$–$V$) behavior (Fig. 2b) of the device. Voltage-dependence of $d_{11}$-SHG signals exhibits qualitatively similar trends at the three excitation wavelengths: it is always below the noise level at low voltage ($V_{DS} < V_t \sim 20$ V, where $V_t$ is the threshold voltage), then increases dramatically when $V_t < V_{DS} < V_s$ (where $V_s \sim 40$ V is the saturation SHG voltage) and eventually saturates at $V_{DS} > V_s$ (Fig. 2a). $d_{11}$-SHG excited by 1018 nm laser shows a higher $V_s$ and saturation SHG ($I_{2\omega,s}$) signal compared to 1043 nm and is expected since the third-order nonlinear coefficient ($\chi_{xxxx}^{(3)}$) is larger near the band edge of CdS (at half the wavelength of FW, 1018 nm). Interestingly, the $I$–$V$ curve exhibits qualitatively similar behavior as the $d_{11}$-SHG signal, with the current increasing and then saturating with the applied voltage (Fig. 2b).

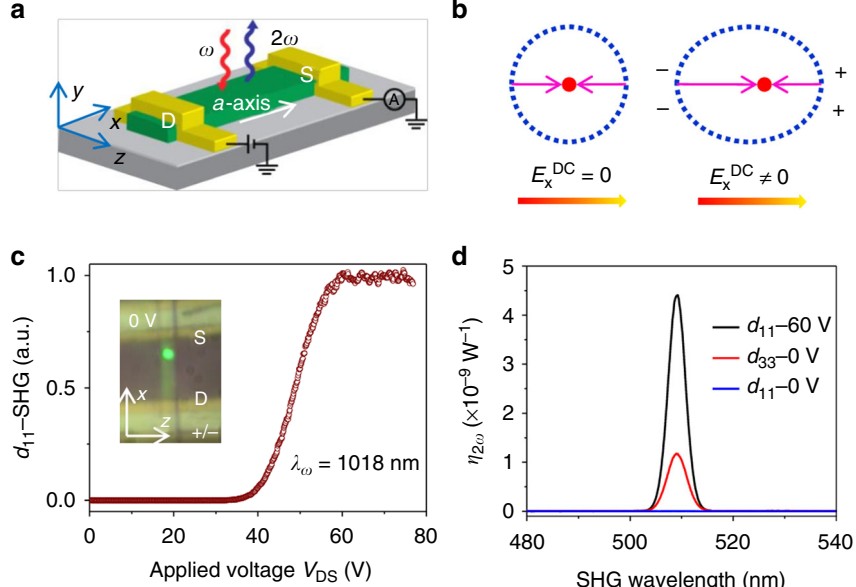

**Fig. 1** Field-induced second-order nonlinear coefficient ($d_{11}$) in a CdS nanobelt. **a** Schematic of a CdS nanobelt device with the source (S) and drain (D) electrodes. The long-axis of the nanobelt ($x$-axis) is determined to be CdS $a$-axis, which is perpendicular to the $z$-axis (or CdS $c$-axis). The fundamental wave at the frequency of $\omega$ is normally incident upon the belt and excites the second-harmonic wave at $2\omega$ which is back-scattered. **b** Schematic of the response of the electron wave function to a strong d.c. applied field. An atom with a positively charged nucleus (red dot) is surrounded by electron cloud (blue dots). If there is no external electric field, the electron wave function is symmetric along the $x$-axis, which can be distorted by an external electric field, thus turning on certain field-induced second-order nonlinear coefficients. **c** Voltage-dependence of $d_{11}$-SHG signal. Inset: device image when the fundamental wave (FW) is incident upon the CdS nanobelt near the source electrode (S). The S-electrode is grounded (0 V) and the drain electrode (D) is biased. When applying a positive bias ($V_{DS} > 0$), the electric current is along the $x$-axis (from D-electrode to S-electrode), with the reverse bias on the S-contact ($V_{DS} > 0$). **d** Normalized conversion efficiency ($\eta_{2\omega} = P_{2\omega}/P_\omega^2$) of second-harmonic generation (SHG) resulting from different nonlinear coefficients ($d_{11}$ and $d_{33}$) at different applied voltages ($V_{DS}$), 0 and 60 V. In order to detect the $d_{11}$-SHG signal, we focused the FW polarized along the $x$-axis ($I_{\omega,x}$) and detected the SHG component polarized along the $x$-axis ($I_{2\omega,x}$) by adjusting the polarizer in front of the detector. Similarly for $d_{33}$-SHG, we employ $I_{\omega,z}$ to detect $I_{2\omega,z}$

The slope of the $I$–$V$ curve increases when $V_{DS} < V_t$, decreases ($V_t < V_{DS} < V_c$) and eventually becomes zero ($V_{DS} > V_c$, where $V_c \sim 35$ V is the saturation current–voltage) (Fig. 2b). Similar $I$–$V$ characteristics are also observed at 1043 nm, but with a smaller saturation current–voltage ($V_c$) and saturation current ($J_s$). Although, higher saturation SHG ($I_{2\omega,s}$) and higher saturation current ($J_s$) are observed at 1018 nm compared to 1043 nm (Fig. 2a, b), the electric field is the dominant cause rather than the current to induce SHG. This is because no SHG is observed (Fig. 2a) while the current increases with voltage ($VDS < V_t$) (Fig. 2b), and the SHG signal increased monotonically (Fig. 2a) while the current remained unchanged between $V_c < V_{DS} < V_s$ (Fig. 2b). Moreover, the corresponding current density (~$10^3$ A cm$^{-2}$) observed in our case is several orders of magnitude lower in comparison to other reports where current density > $10^6$ A cm$^{-2}$ was observed to modify SHG signals[14,15]. Therefore, the saturation of SHG signals at $V_{DS} > V_s$ indicates that a constant electric field is built in the optical excitation region (Fig. 2a).

**High-field domain model for $d_{11}$-SHG.** This $I$–$V$ behavior can be understood to originate from the Schottky junction (Fig. 2b and Supplementary Fig. 2), since one contact is near Ohmic (i.e., Schottky with very small built-in potential), while the other is Schottky. While a typical Schottky barrier can enable observable SHG[9], it is unlikely to induce very-large signals and also saturation of SHG, because the internal electric field changes with applied voltage. However, in semiconductors, one can achieve domains with constant and strong electric fields, called the high-

field domain[16,17]. For example, in materials such as GaAs, the high-field domains can be formed with the electrons scattered into different conduction band valley states accompanied by a decrease in electron mobility with an increase in electric field[16]. In other semiconductors, such as CdS, another proposed mechanism for the high-field domain is due to carrier quenching[17–21]. At high electric fields, trapped holes can be ionized, which can then recombine with electrons in the conduction band, thereby decreasing the overall free-carrier density. This process, described as field quenching[18], can create distinct high and low electric field domains. While the high-field domain has been studied in bulk CdS (thickness > 100 μm) requiring very-high applied voltage (~kV)[20,21], it still remains unclear if the high-field domain can be achieved in nanostructures (~100 nm thickness), and thus its dynamics (e.g., initiation and propagation) and unique attributes, such as their control by external stimuli remain unknown. Additionally, the nonlinear optical properties of high-field domains remain unknown along with their interactions with excitons that can influence bulk nonlinear optical processes.

The stationary high-field domain observed in bulk CdS in earlier studies produces a strong and constant electric field, which spatially expands with increasing voltage[17,19,20]. Once the high-field domain is initiated, the current also starts to saturate[17,19] due to carrier depletion. Due to the similar features (i.e., constant field and current), our observations (SHG and $I$–$V$) are likely related to a stationary high-field domain formation in the CdS nanobelt. In order to validate this hypothesis, we studied the underlying mechanism of the high-field domain in our system and present a model based on field-induced ionization of deep-

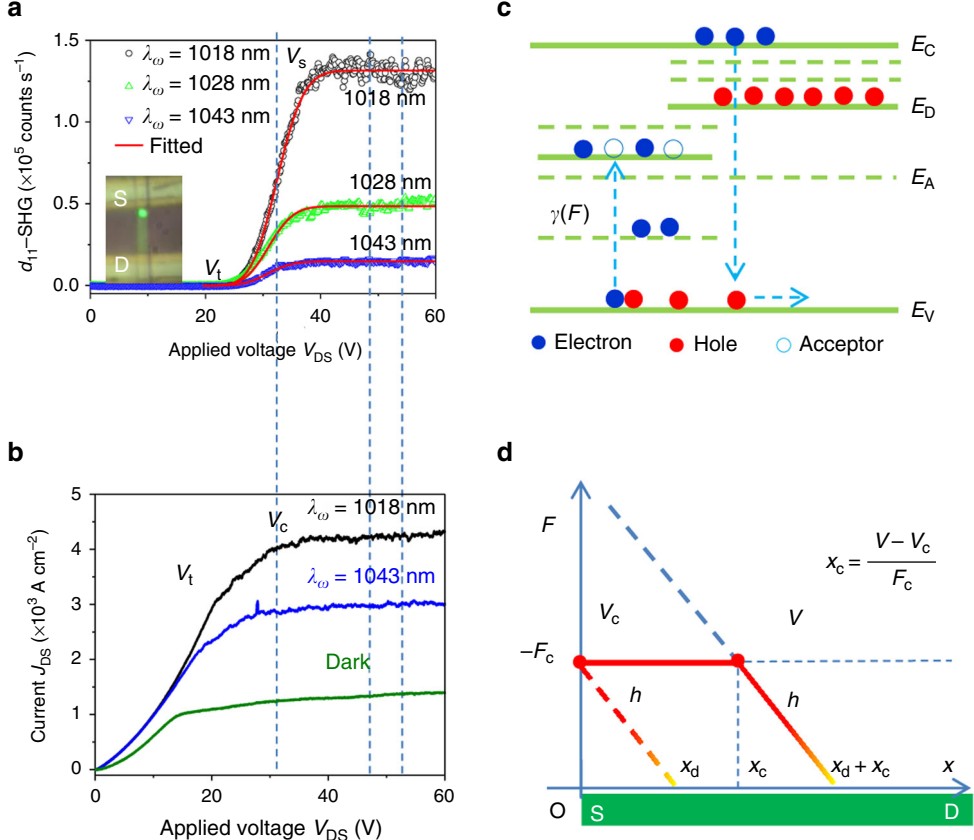

**Fig. 2** Voltage-controlled $d_{11}$-SHG signal and mechanism in CdS nanobelt. **a** $d_{11}$-SHG signal as a function of applied voltage under three excitation wavelengths, 1018, 1028, and 1043 nm. Each curve includes three different regions, $V_{DS} < V_t$, $V_t < V_{DS} < V_s$ and $V_{DS} > V_s$. Here $V_t$ is the turn-on voltage or threshold voltage while $V_s$ represents the saturation voltage where SHG signal begins to saturate. The $d_{11}$-SHG response when $V_t < V_{DS} < V_s$ is fitted by using equation 1 and Supplementary Equation 19 in Supplementary Note 4. **b** $I$–$V$ characteristics of the device in the dark and under excitation at 1018 and 1043 nm. $V_c$ stands for the saturation current–voltage. **c** Schematic of the band structure with donors ($N_D$) and deep deep-level acceptors ($N_A$). Electrons (represented by blue dots) can be excited from the valence band to the accepter traps via a strong field (i.e., field-induced ionization of acceptor traps), to generate holes (represented by red dots), which can then recombine with electrons in the conduction band and also take part in conduction. $\gamma$ is the transition rate driven by the electric field. **d** Schematic of the electric field profile using the Schottky high-field domain model. $F_c$ represents the critical field at which field-induced ionization takes place efficiently when the applied bias $V_{DS}$ equals the critical bias voltage, $V_c$, $h = qN_D/(\varepsilon_r\varepsilon_0)$ and $x_d = F_c/h$. Please note that this is a simplified model by assuming the net space charge density ($\rho$) is changed from $qN_D$ to be zero at the critical electric field

level acceptor traps in a Schottky barrier (i.e., Schottky high-field domain model, Supplementary Note 3). In CdS, the majority of defects are stoichiometric defects, which are formed during synthesis[22] and consist of sulfur ($V_S$) and cadmium ($V_{Cd}$) vacancies and interstitials ($Cd_I$ and $S_I$)[23–28]. CdS is intrinsically $n$-type due to the presence of $V_S$ as donor traps (ionized donor density: $N_D$)[23,24]. $V_{Cd}$ in CdS have also been studied[27] and demonstrated to function as acceptor traps[28], which are initially neutral and become ionized or negatively charged after accepting an electron[29]. An increase in the reverse bias voltage can expand the Schottky barrier along the device and increase the internal electric field. As the internal field approaches a critical value ($F_c$) at $V_c$, the strong electric field can excite electrons from the valence to the acceptor states in the forbidden band-gap region[25–27], which then become ionized and negatively charged (ionized acceptor density: $N_A$)[29]. We refer to this process as field-induced ionization, which reduces the net space charge density ($\rho$) in the Schottky region from $\rho = qN_D$ (below the critical field) to $\rho = q(N_D - N_A)$ at $F_c$ (Supplementary Fig. 3). If $N_D = N_A$, the net space charge is zero ($\rho = 0$) and the internal field remains constant at $F_c$ ($dF/dx \propto \rho = 0$) (Fig. 2d and Supplementary Fig. 3). The region

that develops a constant field is the so-called high-field domain[17,19], followed by a domain boundary and a lower-field region (Fig. 2d). The domain boundary region has a similar field profile as a Schottky barrier, which is formed at $V_{DS} = V_c$, and then moves away from the metal–semiconductor interface if $V_{DS} > V_c$. Further increase of voltage only expands the spatial length of the high-field domain by ionizing more traps in the new region.

Based on the Schottky high-field domain model, we can quantitatively understand the experimental results by calculating the SHG field (Supplementary Note 4)[13],

$$
\begin{aligned}
E_{2\omega}(x_c, x_0) &= \chi^{(3)}_{xxxx} F_c I_\omega \int_0^L f(x, x_c) g(x, x_0, w) dx, \\
&= \chi^{(3)}_{xxxx} F_c I_\omega G(x_c, x_0, w)
\end{aligned}
\tag{1}
$$

where $I_\omega$ is the FW intensity, $x_c = (V - V_c)/F_c$ represents the length of high-field domain, $L$ is the total sample length between the two contacts, $f(x, x_c)$ is the spatial profile of the internal field (Supplementary Equation 10 in Supplementary Note 3) and $g(x, x_0, w)$ stands for the Gaussian beam profile of FW, centered at $x_0$ with a width, $w$ (Supplementary Equation 11 in Supplementary

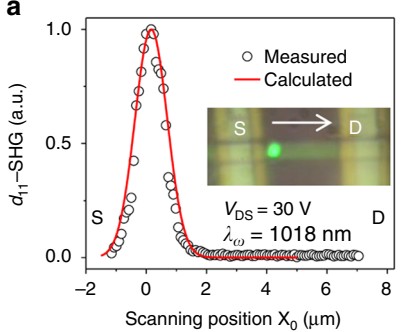

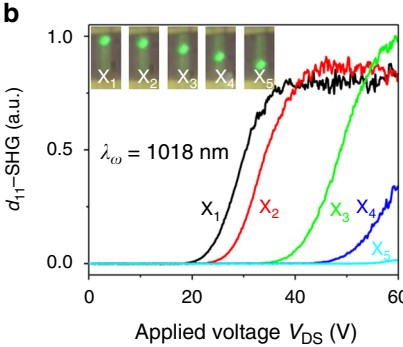

**Fig. 3** Laser position dependence of $d_{11}$-SHG signal along the CdS nanobelt device. **a** $d_{11}$-SHG signal scanned over the entire nanobelt from the source to drain electrodes at $V_{DS} = 30$ V. The calculated curve was obtained using equation 1 and Supplementary Equation 19 in Supplementary Note 4. **b** Position dependence of voltage-controlled SHG. Inset: optical images of the nanobelt device when the FW was incident at different regions at $X_1$, $X_2$, $X_3$, $X_4$, and $X_5$ from source to drain electrodes

Note 4). $G(x_c, x_0, w) = \int_0^L f(x, x_c) g(x, x_0, w) dx$ is the overlap between the Gaussian beam and the region with the spatial field profile given by the function $f(x, x_c)$. Therefore, the SHG signal is $I_{2\omega} = E_{2\omega}^2 = (\chi_{xxxx}^{(3)} F_c I_\omega G)^2$ and exhibits an error function-like behavior (Supplementary Equations 12 and 19 in Supplementary Note 4), which matches the experimentally measured voltage-dependence of $d_{11}$-SHG (Fig. 2a, the red curve fitted to equation 1). Once the high-field domain completely overlaps with the excitation region ($x_c > x_0$), $A$ is constant and the average field equals $F_c$, upon which the SHG signal saturates (Fig. 2a). Therefore, the saturation SHG voltage ($V_s$ at $x_c > x_0 > 0$) is distinct from the saturation current–voltage ($V_c$), describing when the high-field domain is initiated (at $x_c = 0$) with $V_s = V_c + F_c x_0$.

**Position dependence of $d_{11}$-SHG.** In order to further validate the Schottky high-field domain model, we fixed the applied voltage at $V_{DS} = 30$ V in our experiment and measured the $d_{11}$-SHG signal over the entire length of the device (Fig. 3a, inset) by scanning the FW laser. We observe that the $d_{11}$-SHG signal first increases when the laser spot gradually moves onto the CdS region from the metal, and then decays suddenly as the laser is moved away from the metal–CdS interface (S-electrode), where the Schottky barrier is initially formed, as predicted by the Schottky high-field domain model (Eq. 1). To understand the spatial dependence of the high-field domain, we measured SHG at different excitation positions ($X_1$–$X_5$) on the device with the same FW intensity (Fig. 3b). It can be seen that the voltage-controlled $d_{11}$-SHG exhibits qualitatively similar behavior at different positions ($X_1$–$X_5$), but greatly differs in threshold ($V_t$) and saturation voltage ($V_s$) values. For example, at $X_1$ (close to the S-electrode at $x = 0$) where the high-field domain is initiated, the SHG signal saturates at $V_s = 38$ V, while at position $X_2$ and $X_3$, $V_s = 45$ and 60 V, respectively, indicating the high-field domain expands from position $X_1$ to position $X_3$, and eventually to position $X_4$ and $X_5$ with a continued increase of applied voltage. This is consistent with our model, thus demonstrating that the high-field domain due to field-induced ionization enables the switching of otherwise zero second-order nonlinear coefficients and contributes to the unique SHG behavior. The slight differences in the saturation SHG at some positions is likely due to inhomogeneity in the CdS nanobelt (Fig. 3b). Sample inhomogeneities may affect the high-field domain in internal field strength, and hence related output SHG intensity, but will not impact the origin of modulation and mechanism of SHG. These inhomogeneities can be minimized with higher-quality sample growth techniques.

**Fundamental wave power dependence of $d_{11}$-SHG.** While the high-field domain is demonstrated to induce very-strong SHG signals, the ability to further control the high-field domain can broaden its applications for on-chip photonic devices. The SHG signals, intrinsically linked to high-field domains can be manipulated by changing the occupation of the trap states, which can be either activated via near-IR excitation or gate voltage. To study the influence of near-IR excitation on high-field domains, we measured the excitation intensity dependence of $d_{11}$-SHG by tuning the FW at 1018 nm near the D-electrode, under reverse bias, i.e., $V_{DS} < 0$, (Fig. 4a, inset). The normalized SHG conversion efficiency ($\eta_{2\omega} = P_{2\omega}/P_\omega^2$) plotted against applied voltage shows that higher near-IR excitation intensity (or FW) leads to a higher saturation intensity of SHG signals and also saturation voltage ($V_s$) (Fig. 4a). This process is distinct from the conventional SHG process, where the normalized conversion efficiency ($\eta_{2\omega}$) is independent of the excitation intensity[30]. Moreover, higher near-IR intensity also induces higher saturation currents ($J_s$) and saturation current–voltage ($V_c$) (Fig. 4b). We can understand this observation by noting that in CdS there are many available mid-gap donor and acceptor traps[27]. Upon IR excitation, the electrons can be excited from the valence band into mid-gap states (which usually decreases the current)[25], or from mid-gap states into the conduction band (which increases electron density in the conduction band and the overall current)[27]. Typically, IR quenching would dominate, although because of the presence of the high-field domain, most of the low-level acceptor traps are already filled, allowing for more electrons to be excited from trap states (unionized donor and ionized acceptor) into the conduction band, and thereby increase the current (Fig. 4b, Supplementary Fig. 4b and Supplementary Note 5). Since IR excitation increases $N_D$ and decreases $N_A$ (thus, $N_D > N_A$), a higher electric field is required to induce additional $N_A$ (i.e., ionize more acceptor traps) to maintain the condition ($N_D = N_A$) for space charge neutrality (Supplementary Equation 9 in Supplementary Note 3), therefore re-establishing a new high-field domain with a higher critical field that produces stronger SHG (Fig. 4c). Based on this explanation, a shorter excitation wavelength (e.g., 1018 nm) can excite electrons from deeper-level ionized acceptor traps than 1043 nm, thus inducing higher saturation voltage ($V_s$), current ($J_s$), and field ($F_c$), which are consistent with our observations (Fig. 2a, b).

The influence of near-IR excitation on SHG signal intensity can also be observed at the high-field domain boundary (Fig. 4a). At $V_{DS} = -30$ V, the SHG signal does not saturate yet and the near-IR excitation region is partially or completely overlapping with

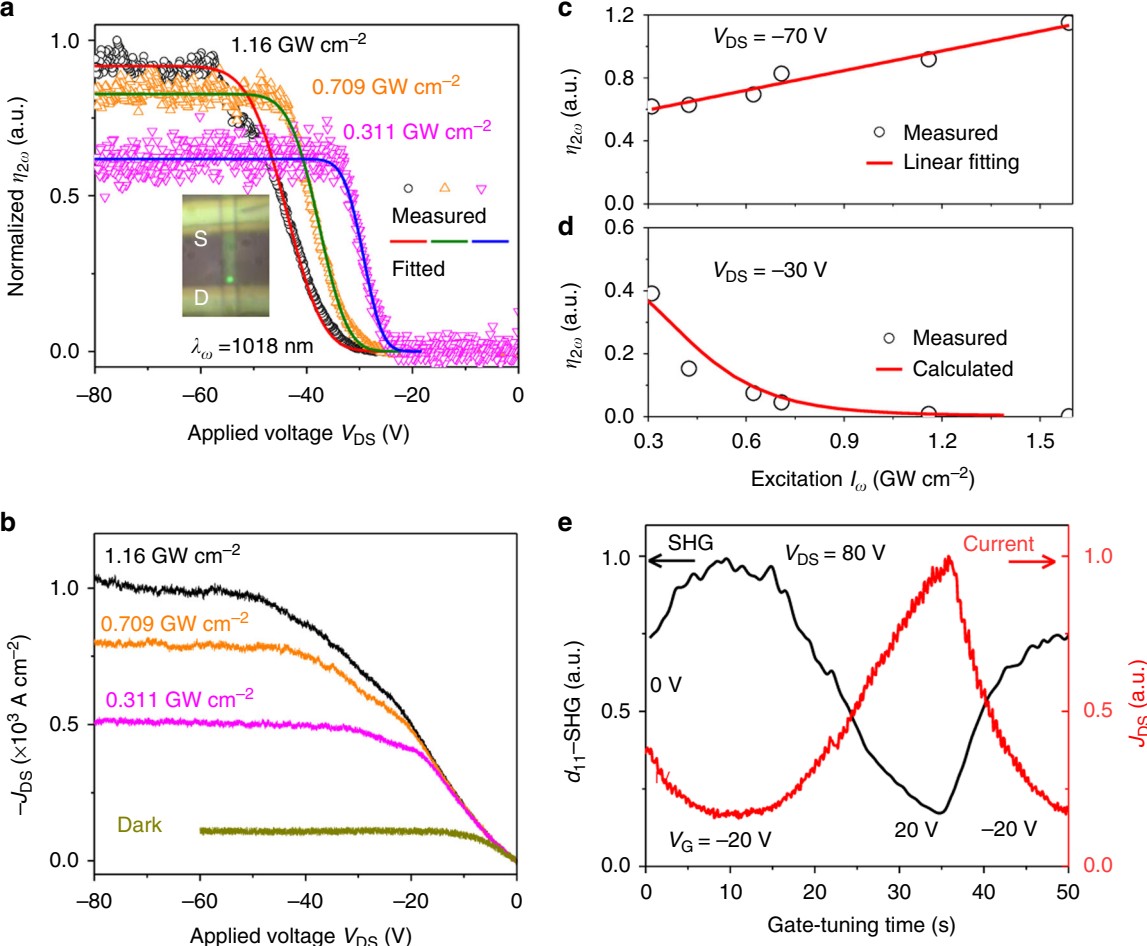

**Fig. 4** Near-IR laser intensity and gate voltage-dependence of $d_{11}$-SHG. **a** Voltage-controlled normalized conversion efficiency ($\eta_\omega = P_\omega/P_\omega^2$) of $d_{11}$-SHG under different excitation intensity of the fundamental wave. **b** I–V curve under different laser excitation intensity. **c** Near-IR intensity dependence of $\eta_{2\omega}$ at −70 V bias, which is within the saturation region or at which the high-field domain overlaps with the IR excitation region completely (supplementary Fig. 4c). **d** IR intensity dependence of $\eta_{2\omega}$ at −30 V bias at which only the domain boundary reaches the near-IR excitation region (Supplementary Fig. 4c). The SHG at -30V was also calculated from the field strength of the high-field domain with the experimental excitation intensity and SHG conversion efficiency. More details can be found in Supplementary Note 5. **e**, Gate-controlled SHG (black curve) and current (red curve). The high-field domain is formed over the near-IR excitation when $V_{DS} = 80$ V (Supplementary Fig. 5b)

the high-field domain boundary (Fig. 4a). If the near-IR laser intensity is increased, the SHG signal ($\eta_{2\omega}$) decreases (e.g., at −30 V, Fig. 4d) showing the opposite trend that was observed earlier inside the high-field domain where the SHG signal increases (e.g., $V_{DS} = -70$ V, Fig. 4c). In addition, the higher near-IR intensity results in a larger saturation voltage ($V_s$ in Fig. 4a) at the same near-IR excitation region, which implies that high near-IR intensity causes a reduction in the spatial length of the high-field domain at a fixed voltage, $V$ ($x_c = (V-V_c)/F_c = X_0 + (V-V_s)/F_c$, Fig. 4a) leading to a poor overlap between the high-field region and the laser spot (centered at $x_0$), generating weaker SHG (Fig. 4d) signals. This is in agreement with our earlier observation that the high-field domain length is proportional to applied voltage (Fig. 3b). These observations demonstrate that the high-field domain can also be controlled via near-IR excitation, which can be useful for their optical manipulation.

**Gate voltage-dependence of $d_{11}$-SHG.** The dynamic control over the SHG signal can be further demonstrated with the application of a gate voltage, which can change the occupation density ($N_D$) of ionized donor traps and consequently change the condition for

initializing the high-field domain ($N_D = N_A$). We observed that when a negative (positive) gate was applied, the current ($J_{DS}$) decreased (increased), whereas SHG within the high-field domain (at $V_{DS} = 80$ V, Supplementary Fig. 5b) showed the opposite response and increased (decreased) (Fig. 4e). Upon application of a positive gate voltage (e.g., $V_G = 20$ V), a large number of electrons are injected from the contact, which can increase the current (red curve, Fig. 4e). Ionized donor traps can be filled and neutralized more easily by accepting electrons, which decreases $N_D$. Therefore, fewer acceptor traps need to be ionized (lower $N_A$) to satisfy $N_D = N_A$, which require lower applied electric fields and hence induce weaker SHG signals (black curve, Fig. 4e). Alternatively, the application of a negative gate voltage (e.g., $V_G = -20$ V) can remove electrons from the traps, causing a decrease in the current (red curve, Fig. 4e) and increase in $N_D$ by ionizing neutral donor traps. Therefore, in order to achieve the high-field domain, more deep-level acceptor traps need to be ionized by a higher electric field to increase $N_A$, which thus generates a stronger SHG signal in comparison with no gate voltage ($V_G = 0$ V). This behavior is in contrast to previously reported gate-controlled SHG owing to gate-tuned charged excitons[31] or gate-induced

accumulation of carriers[32], and demonstrates a completely different mechanism through gate voltage-engineered density of ionized donor traps to control the electric field strength and profile inside the material to generate very-strongly modulated SHG signals that may find applications in high-contrast optical modulator and transistor devices.

## Discussion

We observed $d_{11}$-SHG with an applied voltage with important attributes, such as super-quadratic voltage and power dependence of the nonlinear signal and saturation, and demonstrated the underlying mechanism (i.e., high-field domain) through a range of control measurements. The second-order nonlinear coefficient, $d_{11}$, is intrinsically zero due to the mirror symmetry and can be tuned up to 151 pm V$^{-1}$. This value was obtained by comparing $d_{11}$-SHG with $d_{33}$-SHG signals, which are produced from the same sample and with the same experimental conditions (except fundamental wave polarization). In CdS, $d_{33} = 78$ pm V$^{-1}$ is the largest intrinsic second-order nonlinear coefficient[13] and cannot be tuned when the electric field is applied along the $x$-axis. Analogous to a conventional electrical transistor, the signal ON/OFF ratio of our SHG device is estimated to be >10$^4$. The modulation strength or subthreshold slope is around 200% V$^{-1}$, which is obtained near the threshold voltage (see Supplementary Note 6 and Supplementary Fig. 6). Please note that the saturation and threshold voltages depend on pump conditions, such as fundamental wavelength, spatial location, and power. The response time of the device (see Supplementary Note 7) was estimated to be <1 μs from the density of acceptors ($N_A$), the thermal velocity of the electrons, and the field-enhanced cross-section for capturing electrons[27]. However, the response time can become much faster due to strong applied fields that modify the carrier trapping/de-trapping rates. The response time could also be improved via improved device geometries, such as smaller cross-section and smaller interaction region with the fundamental beam, and also by operating the device close to the threshold region. At high current densities, the sample heating may also influence SHG signals; however, in comparison to the strong field-induced SHG signals, heating does not have a significant impact on $d_{11}$-SHG signals in our case (see Supplementary Note 8).

In conclusion, by applying an external voltage, which breaks certain mirror symmetries, we are able to turn on and modulate the second-order nonlinear coefficient ($d_{11}$), which is otherwise zero in CdS nanobelts. The field-induced $d_{11}$-SHG signal rapidly increases with the applied voltage and eventually saturates, exhibiting a large ON/OFF ratio of ~10$^4$ and a steep subthreshold slope of ~200% V$^{-1}$ and is much larger than previously reported values[10,11]. This unusual SHG behavior is attributed to the high-field domain generated in the CdS nanobelts, which is initiated at the Schottky contact region owing to the field-induced ionization of acceptor traps and can also be controlled by near-IR excitation and source–drain and gate voltage. Further improvement in device properties may be obtained by engineering defects in these materials and one can also envision utilizing high-field domains to induce large optical nonlinearities in materials-like GaAs for similar functionalities. Our study demonstrates a new way to dynamically control nonlinear optical signals in nanoscale materials with ultrahigh signal contrast and signal saturation, which can enable the development of nonlinear optical transistors and modulators for on-chip photonic devices with high-performance metrics and small-form factors, which can be further enhanced by integrating with nanoscale optical cavities[28].

## Methods

**CdS nanobelt growth and device fabrication.** Au-catalyzed single-crystalline CdS nanowires and nanobelts were grown on quartz substrates at 850 °C in a quartz tube with vapor phase transport of CdS powder by argon (99.999% purity, 100 SCCM) and then dry-transferred onto a silicon substrate with a 300 nm SiO$_2$ thermally grown oxide layer. The devices were fabricated by electron beam lithography (Elionix) followed by electron beam evaporation (Kurt Leskter PVD 75) of metal. In order to fabricate Schottky diode devices, lithography and metal deposition were performed for each metal contact in separate steps, with either (200/100/200) Ti/Au/Al or (200/100) Au/Al. No current leakage was observed in the device via the SiO$_2$ layer (source–drain or gate) and also no measurable SHG was observed without the presence of a CdS nanobelt. If the laser was moved away from CdS device, no optical SHG signals were observed. Furthermore, SHG polarimetry data (Supplementary Note 1) matches with crystalline CdS showing that the measured signals were recorded from the CdS.

**Optical and electrical measurement.** The CdS nanobelt devices were mounted in a continuous flow optical microscopy cryostat (ST-500, Janis research) with electrical feedthroughs connected to an electrical measurement system, for room-temperature and low-temperature measurements. The voltage bias was sourced (0–200 V) and the output current signal was converted to an amplified voltage signal by a current preamplifier (DL instruments model 1211) and recorded continuously (~10 data points per second) by PCI card (National Instrument, NI PCI-6281). A femtosecond pulsed Ti: sapphire laser (Chameleon), tuned from 680 to 1080 nm with ~140 fs pulse width and 80 MHz repetition rate, was used to perform SHG measurements. The laser polarization was controlled by a half-wave plate (HWP) and then focused (spot size ~3 μm) onto individual nanobelts by a home-built microscope equipped with a ×60, 0.7 NA objective (Nikon). The back-scattered SHG signals were imaged by a cooled charge-coupled device (CCD) and detected by a spectrometer (Acton) with a 300 groove mm$^{-1}$ 500 nm blaze grating with a CCD detector (Princeton instruments) with a spectral resolution of 0.1 nm.

**Data availability**. The data that support the findings of this study are available from the authors upon reasonable request.

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

## Acknowledgements

This work was supported by the US Army Research Office (grant numbers W911NF-11-1-0024 and W911NF-12-R-0012-03) and NSF-MRSEC (LRSM) seed grant under award number DMR11-20901. Nanofabrication and electron microscopy characterization was carried out at the Singh Center for Nanotechnology at the University of Pennsylvania.

## Author contributions

M.-L.R. and R.A. conceived the concept and designed the devices and experiments. M.-L.R. was responsible for the growth of CdS nanobelts and measurement of SHG signals and developed the high-field domain model. J.B. fabricated and tested CdS devices and performed control measurement. W. L. helped measure SHG from CdS. G.L. performed fitting on experimental results using the high-field domain model. M.-L.R., J.B., and R.A. analyzed the results and wrote the manuscript.

## Additional information

**Competing interests:** The authors declare no competing financial interests.

