## [Peer Review File · Nature Communications]

Reviewers' comments:

Reviewer #1 (Remarks to the Author):

The revised manuscript reads much better. The important findings have been emphasized, misleading terminology has been reduced and many open issues have been answered. The work on an effective nanometric SHG tuning mechanism is generally feasible for publication in Nature Communication but some old and some new issues remain to be addressed before that.

(1) Repeating point 2 of my original review, I believe basic laser parameters such as pulse length, pulse intensity (amplified or not), repetition rate and wavelength range to be too important for a nonlinear optical experiment to shift it into the technical section or supplement. It should be mentioned on page 4 – a single sentence with all these parameters would suffice.

(2) I repeat point 3 of my original review. The authors still write on page 4 that "wurtzite CdS has inversion symmetry along its a axis". This is a *mirror symmetry* whereas the three-dimensional inversion symmetry is broken by the wurtzite structure. The authors should check the manuscript for other misleading terminology like this.

(3) In general, SHG intensity/susceptibility has a pronounced spectral dependence. So when they refer to a value of 151 pm/V, then what is the wavelength at which this value was obtained? Is it the same as the wavelength of the present experiments? If not, any comparison of values or denomination as "largest" would not be justified.

(4) What exactly does the "on/off ratio of $>10^4$ " mean? I did not find an explicit discussion in the text where the authors clearly say what the on and off state represent (which voltage etc.) and what the measured value is that exhibits the 10^4 ratio (SHG intensity?). Where in the data can we see the factor 10^4 ? When explaining this, it would not do to mention it on page 1 but only explain the term pages later. It should be clear from the beginning what is meant, especially when using it in the first paragraph.

(5) What is the "contrast" used in relation to nonlinear coefficients? What are the maximum and minimum values that are needed to derive a "contrast"? What is a "perfect contrast"?

(6) Does a "modulation strength of 200%/V" make sense? If the SHG intensity is zero at zero Voltage, then even for linear increase the change would be infinite for the first volt, 100% for the second volt, 50% for the third volt etc.

(7) When the authors introduce the term "high-field domain" they should explain what exactly they understand as "domain". Certainly not something ferroic, but maybe simply a material phase? Especially a "domain of constant and strong electric field" (page 6) sounds mysterious.

(8) Returning to point 6 of my original review, is desirable that a normalized second-harmonic conversion efficiency *drops to zero* upon *increase* of the illumination intensity as in Fig. 4D? I cannot imagine that this is it technologically desirable. Or is the purpose of Fig. 4D simply to verify the correctness of their proposed mechanism?

(9) I repeat point 7 of my original review. Do phase-matching issues play a role? The authors' answer was that "the thickness is only a few hundred nm and much smaller than the coherence length of about 800 nm". To me, "a few hundred" is at least 300 which is definitely not "much smaller" than 800.

Reviewer #2 (Remarks to the Author):

This work is the revised version of a previously submitted work. The authors have answered more or less adequately to all my queries. I had still one concern related to characterization. In replying to my concerns about material characterization, the authors have decided to revert to literature. I would have liked to see the characterization of the actual material used. The authors could complement this point in finalizing the revision.

The remaining answers are appropriate.

Now for my conclusions. I am always reluctant to let through a paper built on a high number value, or metric as the authors would probably say, here that of the field induced d_{11} coefficient and

related metrics. This opens the door to further works claiming even larger values without much novelty. The origin of the nonlinearity, the high field region induced SHG, sounded rather attractive enough to me to be also put forward. I nevertheless have no real arguments on this fact apart from a purely subjective position. I can nevertheless say that this work has probably reached now the level of quality to be published in Nat. Commun..

Reviewer #3 (Remarks to the Author):

1. The title could be considered to be misleading because the SHG susceptibility is not created by the authors. The observed effect arises from electric-field-induced SHG (EFI-SHG). The improved title is still misleading. SHG coefficients are material properties and thus cannot simply be turned on or off. The term “near perfect contract” is vague and in no way near the ideal infinity.

2. The switching time of this device is still not addressed. Just as Referee #4 states, these EFI-SHG processes are far too slow to meet the demanding frequency range of 10-100 GHz of modulators. The authors responded by saying that the response time is beyond current experimental capabilities, but it should not be difficult to measure it if you can find a waveform generator and a fast oscilloscope. It is this reviewer's opinion that the authors did not answer this important question. Although the authors discuss experimenting heating and other side effects on SHG, an 80 V voltage is applied on a micro-sized device which can impact the device.

3. Most importantly, the maximum susceptibility coefficient of the CdS/SiO₂/Si device achieved through strong EFISH in this paper is only about 151 pm/V. It is still much smaller than the SHG susceptibility coefficients of common nonlinear optical materials, such as GaAs (e.g. $d_{36}=370$ pm/V, Boyd, Nonlinear Optics) even without any EFISH effect. Many monolayers of transition metal dichalcogenides, such as WS₂, reportedly have shown a SHG susceptibility coefficient of 770-4500 pm/V at resonant optical frequencies. The authors emphasize that they observe a large modulation range/strength, but I find that these claims have not be justified. See Reference: C. Janisch, Y. Wang, D. Ma, N. Mehta, A. L. Elías, N. Perea-López, M. Terrones, V. Crespi, and Z. Liu, Sci. Rep. 4, 5530 (2014).

Reviewer #4 (Remarks to the Author):

With interest I have read the resubmitted manuscript as well as the response letter of the authors to the points raised in the four referee reports. One has to acknowledge that the authors were confronted with quite a few questions. They have responded to most of the questions in a satisfactory way and have provided answers up to a level that one can expect at this moment of investigations. I agree with the authors that one cannot expect answers to all questions at this point, for example, to the question about the response time of the device for dynamical control of the EFISH.

This brings me to the question whether the reported results are important enough to warrant publication in the Nature Communications. I believe that this is somewhat a borderline case. Certainly the comprehensive set of data and the demonstrated non-linearity are impressive. On the other hand, the fundamental effect of efficient SHG by the electric field-induced symmetry breaking is not a fundamentally new effect. On the other hand the relation to domains is indeed interesting and deserves more investigation in future.

Taking all these arguments into account, I would be in favor of publication of the strongly improved manuscript.

Reviewer #1 (Remarks to the Author):

The revised manuscript reads much better. The important findings have been emphasized, misleading terminology has been reduced and many open issues have been answered. The work on an effective nanometric SHG tuning mechanism is generally feasible for publication in Nature Communication but some old and some new issues remain to be addressed before that.

(1) Repeating point 2 of my original review, I believe basic laser parameters such as pulse length, pulse intensity (amplified or not), repetition rate and wavelength range to be too important for a nonlinear optical experiment to shift it into the technical section or supplement. It should be mentioned on page 4 – a single sentence with all these parameters would suffice

Response: We have now mentioned the laser parameters on page 4 of the revised manuscript.

(2) I repeat point 3 of my original review. The authors still write on page 4 that "wurtzite CdS has inversion symmetry along its a axis". This is a *mirror symmetry* whereas the three-dimensional inversion symmetry is broken by the wurtzite structure. The authors should check the manuscript for other misleading terminology like this.

Response: We have changed *inversion symmetry* to *mirror symmetry* in the entire manuscript and supplementary information. Thanks.

(3) In general, SHG intensity/susceptibility has a pronounced spectral dependence. So when they refer to a value of 151 pm/V, then what is the wavelength at which this value was obtained? Is it the same as the wavelength of the present experiments? If not, any comparison of values or denomination as "largest" would not be justified.

Response: 1) The pump wavelength is 1018 nm. 2) At the same pump wavelength, we measured d_{11} -SHG and d_{33} -SHG, and then obtained $d_{11}=151$ pm/V from $d_{33}=78$ pm/V. 3) d_{33} exists intrinsically in CdS due to the non-centrosymmetric structure. It is largest among intrinsic nonlinear coefficients of CdS, which is well known for structures with the point group of 6mm. For CdS, $d_{33}=78$ pm/V and other more details are referred to *Nonlinear Optics* by Boyd. We have clarified this in the revised manuscript and mentioned the pump wavelength specifically to remove any confusion.

(4) What exactly does the "on/off ratio of $>10^4$ " mean? I did not find an explicit discussion in the text where the authors clearly say what the on and off state represent (which voltage etc.) and what the measured value is that exhibits the 10^4 ratio (SHG intensity?). Where in the data can we see the factor 10^4 ? When explaining this, it would

not do to mention it on page 1 but only explain the term pages later. It should be clear from the beginning what is meant, especially when using it in the first paragraph.

Response: Ideally, the ON/OFF ratio should be infinite in our case; however this metric is not appropriate although it makes our results appear more encouraging. The maximum value (i.e. SHG_{max}) is the saturation SHG intensity (e.g. at 57 V in Fig. 1C). The minimum value (i.e. SHG_{min}) is the SHG intensity at ~22 V where *SHG starts to appear above the noise floor*. Therefore, the ON/OFF ratio is $SHG_{max}/SHG_{min} > 10^4$. We cannot explain this in the abstract due to space limitations but have explained it clearly on page 5 of the revised manuscript and in section 6 of Supplementary Materials.

The ON/OFF ratio definition is borrowed from the conventional electronic transistor literature, which also show similar $I-V_{gate}$ characteristics, where above a certain gate voltage signal appears above the noise floor and then eventually saturates. Please note, signal saturation (as we observe here for SHG transistor) is a critical component of electronic transistors and we have observed the same with our SHG devices.

(5) What is the "contrast" used in relation to nonlinear coefficients? What are the maximum and minimum values that are needed to derive a "contrast"? What is a "perfect contrast"?

Response: 1) The *contrast* is used to describe how much SHG light is controlled and tuned. It is defined as,

$$Contrast = \frac{SHG_{max} - SHG_{min}}{SHG_{max} + SHG_{min}}$$

2) The maximum value is the saturation SHG while the minimum value should be ideally zero.

3) The perfect contrast is ~1 if SHG_{max} is much larger than SHG_{min} , which is our case ($SHG_{max}/SHG_{min} > 10^4$). We have clarified this in the revised manuscript on page 5.

(6) Does a "modulation strength of 200%/V" make sense? If the SHG intensity is zero at zero Voltage, then even for linear increase the change would be infinite for the first volt, 100% for the second volt, 50% for the third volt etc.

Response: Yes, the *modulation strength* of 200%/V is very reasonable and a conservative estimation in our case.

- 1) The *modulation strength* metric is adapted from the previously reported methodology in the published literature, e.g. Cai *et al*, Science **333**, 1720 (2011) and Seyler *et al*, Nat. Nanotechnol. **10**, 407 (2015). It is the relative change of SHG per volt and is used to evaluate the performance of the device. Cai *et al* showed the modulation strength of SHG is 7%/V in their Science paper. Seyler et al reported the modulation strength of SHG is 10%/V in the Nat.

Nanotechnol paper. The modulation strength is small in their work because the intrinsic and/or background SHG is very large.

- 2) The *modulation strength* is analogous to the subthreshold slope of a conventional electrical transistor. It indicates the rate at which the OFF state is switched to ON state.

In our work, we chose the OFF state at ~22V where SHG starts to appear above the noise floor and increases significantly. Near 22V, we found $\log(I_{2\omega})$ grows with the applied voltage linearly. Please see more details in the figure below. The same figure has been added in Supplementary Materials (i.e. Fig. S6 in page 19). The slope (~200%/V) is then the modulation strength. Please note that our modulation strength is not at 0 V bias.

- 3) Regarding the comment, “If the SHG intensity is zero at zero Voltage, then even for linear increase the change would be infinite for the first volt” The referee’s assumption is incorrect: The reasons are:

- Due to surface SHG, the SHG intensity is generally not exactly zero at zero voltage. The surface SHG occurs in every material. However, it is not detectable in our case.
- The change in SHG would be insignificant compared with surface SHG in the first volt (*i.e.* $d_{surface} \gg d_{induced}$). This is because the electric-field induced SHG is a weak process due to the lower electric field strength and third-order nonlinear coefficient. SHG is,

$$I_{2\omega} \propto (d_{surface} + d_{induced})^2 = (d_{surface} + bV)^2 \approx d_{surface}^2 + 2d_{surface}bV$$

This is what the referee assumed: SHG is linearly proportional to the applied voltage. Then the modulation strength is $2b/d_{surface}$ and will not

rely on the applied voltage. This is the case studied by Cai et al (Science **333**, 1720 (2011)). Therefore their modulation strength was below 10%/V.

- If the change in SHG is significant compared with surface SHG in the 0-1 V range, *i. e.* $d_{surface} \sim d_{induced}$, SHG is,

$$I_{2\omega} \propto (d_{surface} + d_{induced})^2 = (d_{surface} + bV)^2$$

Then the modulation strength is $2b/d_{surface} + 2V(b/d_{surface})^2$. It will not be infinite. This case is similar to our work. The modulation strength is $\sim 200\%/V$. Please note that our modulation strength was calculated where SHG starts to grow significantly.

(7) When the authors introduce the term "high-field domain" they should explain what exactly they understand as "domain". Certainly not something ferroic, but maybe simply a material phase? Especially a "domain of constant and strong electric field" (page 6) sounds mysterious.

Response: The high-field domain is a spatial region of CdS where the electric field is very large and constant. It looks like a Schottky barrier. It is neither a ferroelectric domain nor a material phase.

The *high-field domain* has been reported extensively in Boer's papers and also in reference to the Gunn effect and these papers have been adequately cited in our manuscript along with a detailed discussion.

(8) Returning to point 6 of my original review, is desirable that a normalized second-harmonic conversion efficiency *drops to zero* upon *increase* of the illumination intensity as in Fig. 4D? I cannot imagine that this is it technologically desirable. Or is the purpose of Fig. 4D simply to verify the correctness of their proposed mechanism?

Response:

- 1) Yes, figure 4D confirms that our proposed mechanism is correct as pointed by the referee. This "abnormal" response of the normalized conversion efficiency to pump power was first observed in our work. It can be explained very well and consistent with our proposed mechanism, which confirms that our model captures the right physics.
- 2) The purpose of Fig.4D is to show and analyze the pump power dependence of SHG. An increase in pump power will increase the electric field strength and decrease the domain length by re-distributing carriers. Hence the observation in Fig. 4C is due to the increase of electric field while phenomenon in Fig. 4D is because the domain length is decreased and the spatial overlap between high-field domain and excitation region becomes smaller. The drop in SHG due to increase in pump power can be easily compensated with a slight increase in applied voltage.
- 3) This experiment shows how the high-field domain responds to the pump power and demonstrates one more degree of freedom to control the high-field domain and SHG output.

(9) I repeat point 7 of my original review. Do phase-matching issues play a role? The authors' answer was that "the thickness is only a few hundred nm and much smaller than the coherence length of about 800 nm". To me, "a few hundred" is at least 300 which is definitely not "much smaller" than 800.

Response: *No, phase matching issue is not important in nanoscale materials.* When phase mismatch occurs between the second-harmonic (SH) light and pump light, the SH light is converted from the pump light in the first coherence length and then converted back to the pump light in the second coherence length. This is one cycle and will be repeated if the sample is very long. If the phase matching is satisfied, the SH light will always be converted from the pump light and grow continuously. This is why the phase matching is very important to achieve high conversion efficiency in bulk materials.

But in our work,

- 1) The sample thickness is only few hundred nanometers (~300 nm) and still within one coherence length (~800-900 nm). Therefore, the phase matching issue is not important;
- 2) If the phase matching issue were significant, it only needs to be considered after d_{11} is induced. It will only affect the conversion efficiency of SHG, but cannot induce d_{11} -SHG. We focus on the generation of d_{11} -SHG which is intrinsically zero. It is produced due to the application a voltage instead of the phase matching. Therefore, the phase matching issue is not relevant.

It is very common to ignore the phase matching issue in nanostructures. For example, Cai *et al* did not consider the phase matching issue when studying SHG from a nanoslot (size~100 nm) [Cai *et al*, Science **333**, 1720 (2011)]; Nakayama *et al* ignored this effect even if SHG was produced along a 1.4- μm -long nanowire [Nakayama *et al*, Nature **447**, 1098 (2007)]; Liu *et al* observed SHG from a 1- μm -thick nanobelt and did not draw any conclusion from the phase matching issue [Liu *et al*, Nano Letters **13**,4224 (2013)]. Therefore, phase matching issue is not important in nanostructures.

Reviewer #2 (Remarks to the Author):

This work is the revised version of a previously submitted work. The authors have answered more or less adequately to all my queries. I had still one concern related to characterization. In replying to my concerns about material characterization, the authors have decided to revert to literature. I would have liked to see the characterization of the actual material used. The authors could complement this point in finalizing the revision. The remaining answers are appropriate.

Response: We have presented material characterization in Fig. S1 of Supplementary Materials., relevant to our current work and also given what is well known in the published literature over the last 50 years about CdS films, crystals and nanostructures. Given the transport and optical properties of CdS, the defects (leading to n-type transport) and their nature are consistent with previously published literature including our work on CdS nanostructures in the past 15 years or so. Our work on high-field domains in CdS nanostructures is also consistent with Boer's work on high-field domains due to the well-known defects in CdS. The exact details of the defects and their density in individual samples are not important as mentioned by Boer in his papers, but a general distribution of these defects typically present in CdS samples lead to the generation of high-field domains in these materials. The differences in these defects (distribution) in different samples manifests as differences in voltages at which the high-field domains are formed in CdS.

Our current work is the first demonstration of the novel nonlinear optical properties of these electrically controlled high-field domains in CdS and demonstrates interesting aspects such as large on/off ratios, perfect contrast, modulation strength etc.

Now for my conclusions. I am always reluctant to let through a paper built on a high number value, or metric as the authors would probably say, here that of the field induced d11 coefficient and related metrics. This opens the door to further works claiming even larger values without much novelty. The origin of the nonlinearity, the high field region induced SHG, sounded rather attractive enough to me to be also put forward. I nevertheless have no real arguments on this fact apart from a purely subjective position. I can nevertheless say that this work has probably reached now the level of quality to be published in Nat. Commun..

Response: Thanks for the referee's comments and agree that one has to be careful when reporting metrics. My group has always ensured to the best of our capabilities to report metrics that are on the conservative side. The data as we presented is out in the open and the readers can make their own analyses. We have answered in detail to referee #1 comments about how we arrived at the metrics related to ON/OFF ratio, contrast and modulation strengths and therefore less would be much less confusion now. The data in our work speaks for itself and even without an explicit mention of the values, the novelty of these devices (in our opinion) remains. We have followed the best practices as reported in the literature to make these estimates and all the relevant data have been presented so that the readers can verify these metrics. As mentioned by the referee, the central result here are the novel nonlinear optical properties of field-induced high field

domains in CdS leading to generation of the d_{11} tensor element from zero to very high values (151 pm/V), saturation on the SHG signal observed for the first time, large ON/OFF ratios leading to perfect contrast and large modulation strengths. These unique attributes are due to the engineered high-field domains in CdS and have never been reported. Our work also studies in detail the mechanisms responsible for these effects and a comprehensive model capturing these observations in a self-consistent manner.

Reviewer #3 (Remarks to the Author):

1. The title could be considered to be misleading because the SHG susceptibility is not created by the authors. The observed effect arises from electric-field-induced SHG (EFI-SHG). The improved title is still misleading. SHG coefficients are material properties and thus cannot simply be turned on or off. The term “near perfect contrast” is vague and in no way near the ideal infinity.

Response:

- 1) We have changed the title to, *Strong modulation of second-harmonic generation with near perfect contrast in semiconducting CdS via high-field domain*. We hope the title is now acceptable.
- 2) We have now explained in the revised manuscript how contrast is defined (also see our response to referee #1). The contrast is defined as, $(SHG_{\max} - SHG_{\min}) / (SHG_{\max} + SHG_{\min})$. The maximum value (i.e. SHG_{\max}) is the saturation SHG intensity (e.g. at 57 V in Fig. 1C). The minimum value (i.e. SHG_{\min}) is the SHG intensity at ~22 V where SHG starts to appear above the noise floor. The perfect contrast means its value is 1, not infinity. In our case, the contrast is ~1 as SHG_{\max} / SHG_{\min} is 10^4 , so the contrast is ~1. Ideal contrast is exactly 1.

2. The switching time of this device is still not addressed. Just as Referee #4 states, these EFI-SHG processes are far too slow to meet the demanding frequency range of 10-100 GHz of modulators. The authors responded by saying that the response time is beyond current experimental capabilities, but it should not be difficult to measure it if you can find a waveform generator and a fast oscilloscope. It is this reviewer's opinion that the authors did not answer this important question. Although the authors discuss experimenting heating and other side effects on SHG, an 80 V voltage is applied on a micro-sized device which can impact the device.

Response:

- 1) Regarding the switching time,
 - We did not claim that our device can overpower the demanding frequency range of 10-100 GHz of modulators. These devices may not be used in conventional routers but can have other applications in nonlinear devices. Not every switching device needs to be used in internet routers and satisfy these very high frequency requirements. More importantly, our work is a more scientific advance than a typical engineering/technological advance and we have focused on never previously reported fundamental nonlinear optical properties of high-field domains in semiconductors to make tunable SHG/nonlinear optical devices with very interesting properties. Compared to previously reported work on E-FISH in different materials, our current work mainly demonstrates the high-field domain can be precisely controlled and applied for tuning SHG and presents at least an order of magnitude improvement in metrics (see two examples out of many in Cai *et al*, Science **333**, 1720 (2011) and Seyler *et al*, Nat. Nanotechnol. **10**, 407 (2015). Controlling SHG in nanoscale systems is a relatively new field and the

community is still discovering these effects and characterizing them. We report largely improved properties and not everything can be expected from one manuscript. We hope further progress along these lines is able to shed new insights and materials/systems with improved properties. The switching time is an important parameter. While it was not studied and presented in previous EFISH work (see two examples above and countless others), the switching time in our work would be related to carrier dynamics and deserves further investigation in the future.

2) The purpose of applying up to 80 V bias is to test how SHG behaves and how the high-field domain expands. Its impact on the device is small because the device still worked very well after tuning the bias voltage between -80 V and 80V repeatedly. In fact, a smaller bias (e.g. <40V in Fig. 2A) is applied to tune SHG to the saturation level. If the geometry and device architecture is further improved, we can bring down the required voltage. Furthermore, since this is a field effect, controlled lower doping would lead to lower currents in the device and hence lower required voltage to reach saturation. Improved designs of schottky barriers, separation between electrodes etc can also bring down the required voltage.

3. Most importantly, the maximum susceptibility coefficient of the CdS/SiO₂/Si device achieved through strong EFISH in this paper is only about 151 pm/V. It is still much smaller than the SHG susceptibility coefficients of common nonlinear optical materials, such as GaAs (e.g. $d_{36}=370$ pm/V, Boyd, Nonlinear Optics) even without any EFISH effect. Many monolayers of transition metal dichalcogenides, such as WS₂, reportedly have shown a SHG susceptibility coefficient of 770-4500 pm/V at resonant optical frequencies. The authors emphasize that they observe a large modulation range/strength, but I find that these claims have not been justified. See Reference: C. Janisch, Y. Wang, D. Ma, N. Mehta, A. L. Elías, N. Perea-López, M. Terrones, V. Crespi, and Z. Liu, *Sci. Rep.* 4, 5530 (2014).

Response: We do not understand this comment at all. Our work *does not focus on producing a material with the highest ever nonlinear coefficient* and there are countless other systems which can have larger nonlinearities, but these cannot be tuned with large modulation, contrast etc. We have utilized a conventional semiconductor CdS to engineer field induced nonlinearities via the creation of high-field domains. Our work shows nonlinear properties with *very large tunability (0-151 pm/V), modulation strength (~200%/V), contrast and on/off ratios.*

In contrast to the referee's other comments, we would like to emphasize:

1) GaAs has a large nonlinear coefficient of $d_{36}=370$ pm/V, but the tunability is only 1 pm/V by current [see PRL **108**, 077403 (2012)];

2) WS₂ also has a large nonlinear coefficient, 770-4500 pm/V, but the modulation strength is still <10%/V, which is 20 times smaller than our work.

3) The paper [C. Janisch et al, Sci. Rep. 4, 5530 (2014)] mentioned by the referee still shows a large nonlinear coefficient for WS₂ (~4500 pm/V), but does not present any tunability and control. There is no modulation strength in Janisch's paper for comparison.

Reviewer #4 (Remarks to the Author):

With interest I have read the resubmitted manuscript as well as the response letter of the authors to the points raised in the four referee reports. One has to acknowledge that the authors were confronted with quite a few questions. They have responded to most of the questions in a satisfactory way and have provided answers up to a level that one can expect at this moment of investigations. I agree with the authors that one cannot expect answers to all questions at this point, for example, to the question about the response time of the device for dynamical control of the EFISH. This brings me to the question whether the reported results are important enough to warrant publication in the Nature Communications. I believe that this is somewhat a borderline case. Certainly the comprehensive set of data and the demonstrated non-linearity are impressive. On the other hand, the fundamental effect of efficient SHG by the electric field-induced symmetry breaking is not a fundamentally new effect. On the other hand the relation to domains is indeed interesting and deserves more investigation in future. Taking all these arguments into account, I would be in favor of publication of the strongly improved manuscript.

Response: We appreciate the referee's comment and support. We never claimed that we discovered the E-FISH phenomena and our manuscript cites relevant publications in this area. As remarked by the referee, we have studied for the first time the nonlinear properties of field-controlled high-field domains in a semiconductor (CdS) and observed interesting properties, which in our opinion is a first such demonstration. We believe that further studies utilizing high-field domains (even on other materials such as Si, e.g., via the Gunn effect) can shed new fundamental insights about the optical properties (linear and nonlinear) of these high-field domains that can then be engineered for photonic applications.

- 1) Our current work is mainly demonstrating a novel way (i.e. high-field domain) to control SHG. We are not claiming to create the next generation of modulators to be used in routers that work on quantum confined stark effect in III-V semiconductors. Our work focuses on the interesting and previously unexplored nonlinear optical properties of high-field domains in CdS. We have discussed the high-field domain in great details including its formation, properties and control. The switching time is interesting but beyond the current scope of our work.
- 2) The origin of SHG is still the electric-field-induced symmetry breaking in our work. The novelty and significance of the SHG output, like saturation and abnormal response to pump power, is due to the high-field domain. This was never observed in previous literatures.
- 3) We have studied the high-field domain extensively in different aspects, like formation, property and control, and presented one significant application in SHG. We are planning to extend this study in different materials and applications and by applying different stimulus.

In response to comments by all the referees, many changes have been made and we hope our revised manuscript can be considered appropriate for publication.

Reviewers' comments:

Reviewer #1 (Remarks to the Author):

I am mostly happy with the renewed revisions. They improve the clarity of the paper and are helpful for non-specialist readers. I only have two final remarks. First, a bracket is missing in the contrast definition on page 5. Second, they should give a number supporting their argument that phase-matching effects are neglected. I understand their argument and that it is common to neglect the effect, but even at $3/8$ of the coherence length, there will be a contribution, presumably in the range of a few percent. So a sentence that phase matching effects are neglected because they do not exceed $??\%$ in the present case will be useful.

Reviewer #3 (Remarks to the Author):

The revised manuscript has been improved and misleading terminology has been reduced, but I still have the same three points to raise about the manuscript: 1. The title is misleading. 2. The switching time is not addressed. 3. The strong modulation (range/strength) of SHG claim is not justified.

In the updated title, SHG "with near perfect contrast" is still misleading. The authors define the SHG contrast as, $SHG\ Contrast = (SHG@57V - SHG@22V) / (SHG@57V + SHG@22V)$. This describes the peculiar situation of the specific device and cannot be generalized to other situations. Based on this, the perfect SHG contrast = 1 when $SHG@22V = 0$.

I would use a more general equation to define the SHG contrast as, $SHG\ Contrast = (SHG@57V - SHG@0V) / SHG@0V$. This definition has been well accepted in optical imaging, where contrast is defined as the difference in light intensity between the image and the adjacent background relative to the overall background intensity. Based on this, the perfect SHG contrast = infinity when $SHG@22V = 0$.

There are still some misleading terms in the manuscript. For example, the pump wavelength should be referred as the fundamental wavelength. When you say "pump", you tend to follow with "probe". When you say "second harmonics", you tend to use the "fundamental" to describe the excitation wavelength.

As the authors mentioned, the switching time is an important parameter, but they have not studied it. The authors must measure or estimate the switching time scales. It should not be difficult to measure the response time of SHG using an oscilloscope when an instant bias pulse is applied on the device.

I also have a new concern. In the high-field domain, the d11-SHG can be tuned from 0-151 pm/V by an electric field, but the d33-SHG stays at a constant 78 pm/V. This sounds not right. I guess d33-SHG should also be tuned by the field.

To realize tuning SHG with a large modulation/contrast, using an electric field is just one way of doing it. A simple way of doing it is to vary the incident laser power on a material with a large nonlinear coefficient. This can also be achieved by mechanically rotating a nonlinear optical crystal. In the response letter, the authors mentioned that SHG can be tuned by electric current in GaAs and by an electric field in WS₂, all with a smaller tunability. However, I believe these effects have not been systematically studied and it is hard to compare the high-field domain effects in these materials to that in CdS.

In conclusion, I think the high field domain-induced SHG has the potential to make a contribution to the field of optical SHG and light-matter interaction in semiconductors and solar energy materials; however, the authors' claim of a large SHG modulation/contrast in CdS has not been justified based on the values reported in this paper. Therefore, I do not recommend the manuscript for publication in Nature Communications in its present form.

Reviewer #1 (Remarks to the Author):

I am mostly happy with the renewed revisions. They improve the clarity of the paper and are helpful for non-specialist readers. I only have two final remarks. First, a bracket is missing in the contrast definition on page 5. Second, they should give a number supporting their argument that phase-matching effects are neglected. I understand their argument and that it is common to neglect the effect, but even at 3/8 of the coherence length, there will be a contribution, presumably in the range of a few percent. So a sentence that phase matching effects are neglected because they do not exceed ???% in the present case will be useful.

Response:

- 1) The missing bracket has been added in the revised manuscript;
- 2) As stated, the phase mismatch should be considered if the sample length is larger than the coherence length. In our current work,
 - This phase mismatch effect was not considered in the theoretical model because it is not the reason for inducing SHG. It only contributes to the signal intensity level after SHG was induced.
 - This effect it is not significant in our nano-devices because:
 - The sample thickness is only few hundred nanometers ($l < 300$ nm) and well within one coherence length ($l_c \sim 800-900$ nm). The SHG field generated in our case is $\sim 5.5\%$ lower than the assumed case of perfect phase matching. Hence the phase mismatch effect is not very significant.
 - The effective nonlinear coefficient, d_{11} , is obtained by comparing d_{11} -SHG with d_{33} -SHG. The factor of phase matching effect is cancelled out (i.e. d_{11} -SHG/ d_{33} -SHG) and has no impact on induced d_{11} .
 - The voltage dependence of SHG is normalized and fitted to the model. The factor of phase matching effect will not have impact on the behavior or trend.

Hence the phase mismatch effect can be ignored given its little impact on our reported values.

We have now added detailed discussion in Section 9 in supplementary information on page 15 clarifying further that phase mismatch effects are not significant and produces only $\sim 5\%$ lower signals in comparison to perfect phase matching conditions.

Reviewer #3 (Remarks to the Author):

The revised manuscript has been improved and misleading terminology has been reduced, but I still have the same three points to raise about the manuscript: 1. The title is misleading. 2. The switching time is not addressed. 3. The strong modulation (range/strength) of SHG claim is not justified.

Response:

We have changed the title to “Strong modulation of second-harmonic generation with very large contrast in semiconducting CdS via high-field domain”. If the referee has issues even with this title, then it may be best if the referee can make constructive suggestions. There are other referees also who all have agreed on the title. But of course, we are open to suggestions and hopefully they are specific.

In the updated title, SHG “with near perfect contrast” is still misleading. The authors define the SHG contrast as, $SHG\ Contrast = (SHG@57V - SHG@22V) / (SHG@57V + SHG@22V)$. This describes the peculiar situation of the specific device and cannot be generalized to other situations. Based on this, the perfect SHG contrast = 1 when $SHG@22V=0$. I would use a more general equation to define the SHG contrast as, $SHG\ Contrast = (SHG@57V - SHG@0V) / SHG@0V$. This definition has been well accepted in optical imaging, where contrast is defined as the difference in light intensity between the image and the adjacent background relative to the overall background intensity. Based on this, the perfect SHG contrast = infinity when $SHG@22V=0$.

Response:

The title has been modified and there is sufficient discussion in the manuscript about contrast ratios and other metrics. Also, what we have is not a peculiar situation. There are many systems that go from zero to a finite value and there are proper methods to quantify these changes rather than simply saying the ratio is infinite. One can easily say that the modulation is infinite as the referee suggests, but it is neither meaningful nor helpful. Electronic transistors also have infinite modulation but nobody says that the transistor sub threshold slope is infinite. We already added a proper scientific figure in SI (supplementary Fig. S6), which shows the accepted technique for electronic transistor devices. And now we have included a very standard and well-regarded reference on electronic transistors. Also, we are not reporting optical imaging data as erroneously mentioned by the referee.

We would like to point out again that if the ON/OFF ratio is very large (i.e. approaches infinity), then modulation contrast becomes more meaningful. It is defined as $(SHG_{max} - SHG_{min}) / (SHG_{max} + SHG_{min})$. If $SHG_{max} \gg SHG_{min}$, the modulation contrast approaches 1, which is the case of perfect contrast. If $SHG_{max} \sim SHG_{min}$, the modulation contrast approaches 0 and no modulation is achieved.

The *contrast* metric suggested by the referee is the *relative change* in SHG intensity, i.e., $(SHG_{max} - SHG_{0V}) / SHG_{min} = SHG_{max} / SHG_{0V} - 1$ and it produces a meaningless number because SHG_{0V} is zero. This is where contrast and modulation strength metrics become important. Modulation strength is defined as relative change in SHG intensity per volt. Our modulation is exceptionally high. The referees can easily see that the signal goes from noise floor to very high values related to changes in d_{11} from 0 to 150 pm/V.

In our revised manuscript, we have further clarified ON/OFF ratio, modulation strength and modulation contrast in Section 6 of supplementary information.

There are still some misleading terms in the manuscript. For example, the pump wavelength should be referred as the fundamental wavelength. When you say “pump”, you tend to follow with “probe”. When you say “second harmonics”, you tend to use the “fundamental” to describe the excitation wavelength.

Response: We have revised “pump wavelength” as “fundamental wavelength” in the revised manuscript.

As the authors mentioned, the switching time is an important parameter, but they have not studied it. The authors must measure or estimate the switching time scales. It should not be difficult to measure the response time of SHG using an oscilloscope when an instant bias pulse is applied on the device.

Response: We have estimated our switching speed and include in our revised m/s a discussion on this topic. Other referees also agree that not everything can be done in one m/s especially when these results are very new and previously unreported. We currently do not have the expertise to measure fast signals in our lab for nanoscale structures and these are not as easy as suggested by the referee as these measurements are on nanostructures utilizing microscopy. Our paper does not claim extremely fast switching for internet applications as addressed in our previous revision and accepted by referees.

To estimate the response time of device, we look at field-enhanced ionization of traps which is responsible for the high-field domain. This is the process of driving electron transition from the valence band to acceptor traps (i.e. deep-level Cd vacancies) via electric field. Note that the cross section of capturing holes is $\sim 10^{-17} \text{ cm}^2$ for these acceptor holes [*J. Appl. Phys.* **52**, 261-268 (1981)]. In order to achieve high-field domain, the large cross section of capturing electrons ($> 10^{-17} - 10^{-18} \text{ cm}^2$) is required to ionize these acceptor traps and can be achieved by applying electric field. Considering the density of acceptor traps $N_A \sim 10^{16} - 10^{17} \text{ cm}^{-3}$, thermal velocity of electron $\langle v \rangle \sim 10^7 \text{ cm s}^{-1}$ and the cross section of capturing electrons $S_c > 10^{-17} \text{ cm}^2$, the response time of acceptor traps can be estimated as,

$$\tau = \frac{1}{S_c N_A \langle v \rangle} \ll 1 \mu\text{s}$$

However, this estimation can become significantly less, leading to a much faster response from the device due to strong applied fields which strongly modulates carrier trapping/de-trapping rates. Also, improved device geometries, smaller cross-section, operating close to the threshold via applying a d.c voltage below the threshold and a modulating voltage to cross the threshold, better spatial overlap, smaller spot size can significantly improve the switching times.

We have added this part in the Discussion part of manuscript and Section 7 in supplementary information.

I also have a new concern. In the high-field domain, the d11-SHG can be tuned from 0-151 pm/V by an electric field, but the d33-SHG stays at a constant 78 pm/V. This sounds not right. I guess d33-SHG should also be tuned by the field.

Response: Since the electric field was applied along the x direction and not z (x =1, z = 3) the question of d₃₃ modulation does not arise from plain symmetry arguments. Therefore, d₃₃-SHG was not tuned in our case. This is now mentioned in the discussion part of the revised manuscript and Section 1 in supplementary information. Also, d₃₃=78 pm/V is an intrinsic nonlinear coefficient and is well known as also reported in *Nonlinear Optics* book by Boyd.

To realize tuning SHG with a large modulation/contrast, using an electric field is just one way of doing it. An simple way of doing it is to vary the incident laser power on a material with a large

nonlinear coefficient. This can also be achieved by mechanically rotating a nonlinear optical crystal. In the response letter, the authors mentioned that SHG can be tuned by electric current in GaAs and by an electric field in WS₂, all with a smaller tunability. However, I believe these effects have not been systemically studied and it is hard to compare the high-field domain effects in these materials to that in CdS

Response:

- 1) An equivalent argument based on the above comments would be that an electronic transistor is not important as one can modulate the current flowing through the device by simply modulating the source-drain bias. If that were true, invention of the electronic transistor would not be given the Nobel prize! Any device or light source can be (trivially) turned on/off by modulating the applied bias or incident laser power that itself powers the device, but it has almost no useful properties. Any useful “switch” therefore has to be controlled by another stimulus (e.g., gate electrode in transistor, another optical excitation source for an all-optical switch etc) and not by the very stimulus that drives the current/light etc in that switch. Therefore, we believe the referee may have missed the entire point of the paper and the functioning principles behind these class of devices that are termed switches, modulators etc. Mechanical rotations of the crystal can also modulate but one can easily imagine these will be extremely slow and completely useless for any on-chip applications.
- 2) In our previous response letter responding to the systems brought to discussion by the referee, our point was that GaAs and WS₂ have large *intrinsic* nonlinear coefficients but the focus of our work is on the ***modulation strength*** instead of their intrinsic value. The current effect modulation in GaAs has been systematically studied by Ruzicka *et al* [*Phys. Rev. Lett.* **108**, 077403 (2012)] and showed a very small modulation. The electric field in WS₂ was also reported by Seyler *et al.* with a very limited modulation (~10%/V) [*Nat. Nanotechnol.* **10**, 407-411 (2015)]. These metrics do not even come close to our observations utilizing the high-field domain based optical nonlinearities. We have compared the range of tunability reported under different mechanisms and show the performance of our device.

The nonlinear optical properties of high-field domains at this stage cannot be compared between these materials as they are not yet reported in GaAs and WS₂ yet. But it would be very interesting to study nonlinear properties of high-field domains in other materials, something that have never been reported before and we hope our work stimulates more activity in this area. We present the first such case for CdS and it should be extended to other systems. A sentence highlighting this aspect has been included in the revised manuscript.

In conclusion, I think the high field domain-induced SHG has the potential to make a contribution to the field of optical SHG and light-matter interaction in semiconductors and solar energy materials; however, the authors' claim of a large SHG modulation/contrast in CdS has not been justified based on the values reported in this paper. Therefore, I do not recommend the manuscript for publication in Nature Communications in its present form.

Response: We disagree with the referee and don't understand why the referee says that a large SHG modulation/contrast in CdS has not been justified based on the values reported in this paper. Actually,

- 1) The value of new d_{11} ($\sim 150\text{pm/V}$) was obtained from our measurements and by a very scientific analysis/methodology. We compared d_{11} -SHG with d_{33} -SHG which was from the same sample and under the same excitation conditions (i.e. wavelength, power, system etc) and derived d_{11} from d_{33} . Compared with other methods using different materials, our method is more reasonable and accurate. The estimation on d_{11} has been described clearly in our paper.
- 2) The modulation strength is defined as the relative change in SHG intensity per volt and widely used in electric field induced SHG [e.g., *Science* **333**, 1720-1723 (2011)]. It is shown to be $\sim 200\%/V$ in our work and is 10 times larger than these reported values [e.g., *Science* **333**, 1720-1723 (2011)].

REVIEWERS' COMMENTS:

Reviewer #3 (Remarks to the Author):

I have read the authors' response and the revised manuscript. The manuscript has been improved. The new manuscript improved the clarity and accessibility to the general reader. The authors have reasonably addressed my earlier specific concerns and I recommend that the manuscript be published in Nature Communications.

Minor points: In the introduction paragraph, "break structural mirror symmetry" should be "break structural inversion symmetry". The authors sometimes use the phrase "high-field domains" but in other places use the phrase "high field domains." This sentence need to be edited: "A femtosecond pulsed Ti: sapphire laser, tuned from 680 nm to 1080 nm with ~140 fs pulse width and 80 MHz repetition rate and focused to a spot size of ~3 μm was used to perform SHG measurements."